# Development and Degeneration of the Intervertebral Disc—Insights from Across Species

**DOI:** 10.3390/vetsci10090540

**Published:** 2023-08-24

**Authors:** Kathryn Murphy, Thomas Lufkin, Petra Kraus

**Affiliations:** Department of Biology, Clarkson University, Potsdam, NY 13699, USA; murphykg@clarkson.edu

**Keywords:** annulus fibrosus, biomarkers, development, intervertebral disc, intervertebral disc degeneration, inflammation, notochord, nucleus pulposus, species

## Abstract

**Simple Summary:**

The intervertebral disc is an important organ providing structure, support and flexibility to the spine, yet it can degrade over an individual’s lifetime resulting in a painful condition known as intervertebral disc degeneration. Historically, the degeneration or breakdown of the organ has been catalogued and studied in humans, mice and to some extent dogs, however research has expanded to rats, cows, horses, rabbits, cats, and monkeys in recent decades allowing for the application of new research methods. Expanded research has further clarified the mechanisms contributing to degeneration at a molecular and cellular level. This review examines stressors promoting degeneration, how the intervertebral disc responds to them, the variation in symptomatic presentation of intervertebral disc degeneration between species, physical differences in the disc at different levels in the spine and between animals, as well as how the cellular population of the disc changes over time. Examining these aspects both within and between species helps to characterize degenerative changes in the intervertebral disc necessary for development of treatment and therapy.

**Abstract:**

Back pain caused by intervertebral disc (IVD) degeneration has a major socio-economic impact in humans, yet historically has received minimal attention in species other than humans, mice and dogs. However, a general growing interest in this unique organ prompted the expansion of IVD research in rats, rabbits, cats, horses, monkeys, and cows, further illuminating the complex nature of the organ in both healthy and degenerative states. Application of recent biotechnological advancements, including single cell RNA sequencing and complex data analysis methods has begun to explain the shifting inflammatory signaling, variation in cellular subpopulations, differential gene expression, mechanical loading, and metabolic stresses which contribute to age and stress related degeneration of the IVD. This increase in IVD research across species introduces a need for chronicling IVD advancements and tissue biomarkers both within and between species. Here we provide a comprehensive review of recent single cell RNA sequencing data alongside existing case reports and histo/morphological data to highlight the cellular complexity and metabolic challenges of this unique organ that is of structural importance for all vertebrates.

## 1. Introduction

Intervertebral discs (IVD) sit between the vertebrae, providing shock absorption and enabling flexibility of the spine. The IVD is made up of three tissue types: cartilage endplate (CEP), annulus fibrosus (AF), and nucleus pulposus (NP), each of which have unique physical and biomolecular properties (Figure 1) [1,2,3]. IVD degeneration (IVDD), which often leads to back pain, osteoarthritis (OA), neuropathy, endplate defects, and disc herniations is categorized by the breakdown of the extracellular matrix (ECM) and changes in the microenvironment of the IVD tissue [2,4]. IVDD is often accompanied by protrusion or extrusion of the IVD into the spinal canal, pushing on nerve roots and the spinal cord which can induce neurologic symptoms [5,6]. Inflammation, changes in cellular function including senescence, and loss of cells through regulated cell death all contribute to degeneration of the NP, AF, and calcification of the CEP and NP [2,7,8,9,10]. IVDD is a common disease in humans, and also affects a large number of other species including monkeys, cows, horses, mice, rabbits, rats, dogs, cats, and bears [11,12,13,14,15]. To date, there are no treatments that effectively halt or reverse IVDD and the changing microenvironment results in further degeneration, innervation, and neovascularization of the disc throughout an individual’s life [2,6,16,17]. Analyzing animal models of IVDD has furthered our understanding of disease onset and progression, yet further research into markers of IVD tissue types, disease progression, and eventual therapy and treatment development is necessary for alleviating degeneration and subsequent pain across species.

## 2. A General Understanding of Anatomic and Molecular Features of the IVD

### 2.1. Importance of the Notochord

Mouse fate-mapping and lineage tracing experiments demonstrated that the NP is a notochord (NC) derived tissue [3,25,26]. As such, the NP contains both NC-like cells and smaller chondrocyte (CC)-like cells contributing to a heterogenous cell population [14,27,28,29,30] with NC-like cells declining or disappearing in the matured IVD of some animals (Figure 2) [14,27,31]. The NC also acts as an important transient structure influencing patterning of ventrolateral sclerotome cells during embryogenesis, which migrate on both sides of the embryonic midline to surround the NC with the axial sclerotome giving rise to AF cells of the IVD and portions of the vertebral body (Figure 1) [9,29,32,33]. Failure to form the NC sheath during development resulted in abnormal NP cell movement and a malformed NP, further illustrating the importance of the NC and NC sheath for NP development [34]. Notably, after the NC sheath disappears during development, NC-like cells remain present in the future NP [3,26,35]. Single cell RNA sequencing (scRNA seq) has recently been instrumental in identifying these clusters of cells, further advancing the understanding of NC cells, NC-like cells, NP progenitors, and NP cells which is further discussed in Section 2.5. Beyond influencing NP and AF cell fate, NC cells also contribute to ECM synthesis through the production of proteoglycans (PG), most importantly aggrecan (Acan), and collagens such as collagen II [9]. Pathway analysis of NC markers suggest a role in the inhibition of inflammation and vascularization [36].

### 2.2. IVD Structure: Nucleus Pulposus

Whether NP cells transdifferentiate from NC-like cells over time or undergo cell death and replacement by CC-like NP cells has been debated [37] (Figure 3). Another study speculated a combined NP cell origin with some cells developing from NC cells and others from CC cells [17]. Recent scRNA seq data has identified NP progenitors with NC origins populating the IVD and contributing to the heterogeneous tissue [35,38,39].

**Figure 2 vetsci-10-00540-f002:**
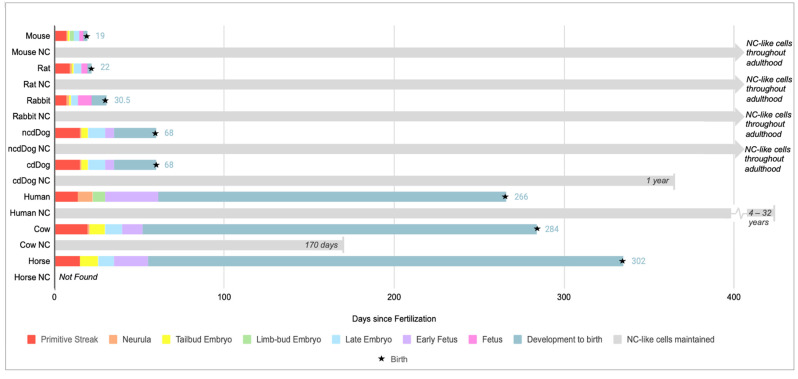
Illustrates the timeline of embryonic development from the onset of gastrulation to birth across different species such as mice, rats, rabbits, non-chondrodystrophic (NCD) dogs, chondrodystrophic (CD) dogs, humans, cows, and horses, in addition to the reported disappearance of notochord (NC)-like cells in the NP. Embryonic development data in humans is compiled from [40] and all other animals from [41] while NC-like cell disappearance is compiled from mice [37], rats [26,42], rabbits [26,42], dogs [26,37], humans [30,42,43], cows (personal observation) and [14], horses [11]. Information on NC or NC-like cells disappearance varied across sources and was based on histological or morphological data. Advancements in scRNA seq analysis can provide further clarification and identified NC-like cells into adulthood in humans.

Compositionally, the ECM of the NP is primarily made up of Acan, type I and type II collagen, as well as hyaluronic acid, creating a hydrated matrix with a high internal osmotic pressure that enables the spine to withstand loading and avoid disc extrusion [2,44,45]. The balance of ECM molecules is important in maintaining a healthy shock absorbing IVD. Unsurprisingly, ECM remodeling occurs with degeneration. In healthy discs, the NP typically has a higher ratio of type II collagen to type I collagen [2]. This collagen ratio shifts to favor type I collagen during degeneration [10,46]. Additionally, during IVDD there is a decrease in Acan and versican (Vcan) and an increase in smaller PGs [9,47]. Negatively charged glycosaminoglycan chains of Acan, interact with cations and water molecules, thus affecting the balance of cations and anions and osmotic pressure within the IVD [48]. However, during degeneration and ECM remodeling the IVD experiences calcification [10] and has diminished osmotic pressure and disc hydration due to a reduction in large PG content and thus fewer interactions with water molecules disrupting the molecular flow in and out of the IVD ECM [46].

The predominantly avascular nature of the IVD contributes to its uniquely harsh microenvironment. Fournier et al. performed a scoping review analyzing vasculature in IVDs from individuals with no history of back pain, radiculopathy, or myelopathy across studies conducted between 1959 to 2018 [17]. Across the studies included, the vast majority agreed that the NP from fetal to infant years is avascular with only one reporting the presence of blood vessels in degenerated discs at the NP and AF border [17,49]. One additional study found microscopic angiogenesis in 6.6% of their human samples aged 2–25 years old and no study reported vascularization in discs from individuals after 25 years old [17]. Consensus supports the NP tissue as avascular throughout an individuals’ lifespan for those with no medical history of back pain. Notably, SOX10, CTSK, and TBXT positive (+) cells in the developing NC/NP tend to express SEMA3A, which is a protein thought to perpetuate avascular environments and therefore may contribute to the NP’s avascular nature [50].

### 2.3. IVD Structure: Annulus Fibrosus

The AF is traditionally differentiated into an outer AF (oAF) and inner AF (iAF) or transition zone (TZ), which is described as the region between the oAF and the NP [14]. The oAF is primarily composed of organized type I collagen layers and elastin while the TZ features a comparatively lower ratio of type I/II collagen as the disc shifts towards NP tissue [2,29,45] (Figure 3). Fibroblast-like cells are responsible for the production of type I collagen while fibrocartilage cells produce type II collagen of the TZ [2,29]. The AF is subject to degeneration characterized by annular fissures as well as biomolecular changes including the disappearance of decorin (Dcn) and biglycan (Bgn) [10,51]. These two small leucine-rich PGs are thought to aid in resistance to biomechanical stress in the outer AF, and their disappearance is reported by fifty years of age in the human IVD [10,51]. Sharpey’s fibers, made primarily of collagen bundles that connect the iAF to the CEP, peripherally surround the NP tissue forming an enclosure that connects the IVD to the VB, securing the IVD’s position in the spine [10,23]. Interestingly, loading a poorly hydrated disc with a lowered osmotic pressure reportedly increased discogenic pain and greater stress on the AF and CEP compared to the loading of a healthy disc [52]. This indicates degenerative changes in one tissue can indirectly stress the remaining tissues thus further degenerating the disc at large.

Unlike the NP, the oAF exhibits some level of vascularization even in a healthy disc. During development, eight of ten IVD studies between 0–2 years reported vasculature in the AF, however it was restricted to the oAF in non-degenerated discs even in aged adults [17]. In the latter group vascularization of the AF was primarily reported in conjunction with fissures or nerve growth in damaged tissue and deeper penetration in highly degenerated tissues [17].

### 2.4. IVD Structure: Cartilage Endplates

The CEP defines the upper and lower boundaries of the IVD, connecting the NP and AF regions to bony vertebrae [53]. The proximity of the CEP to the epiphyseal arteries allows blood supply to this tissue and promotes diffusion of nutrients to the AF and NP [2,10,53]. Of nine studies examining IVDs from patients with no history of back pain conducted between 1947 and 2007, the majority agreed the CEP has some vasculature throughout embryonic development and infancy before transitioning to an avascular tissue by ages 8–10 with a regression in the amount of vasculature in the CEP over time [17]. CEP degeneration, marked by endplate irregularity, more easily allows for NP herniation into the vertebral body creating ‘Schmorl’s nodes’ which can be documented through magnetic resonance imaging (MRI) [4,5]. Schmorl’s nodes were originally described in 1927 in humans as “Knorpelknötchen” and linked to spondylitis [4]. Additionally, four studies reported vascular ingrowth in the CEP of damaged IVDs between ages 2–65+, indicating degenerative changes to the IVD may be accompanied by vascularization [17]. Furthermore, thickening and increased irregularity of CEPs is a marked degenerative change to the IVD [5].

**Figure 3 vetsci-10-00540-f003:**
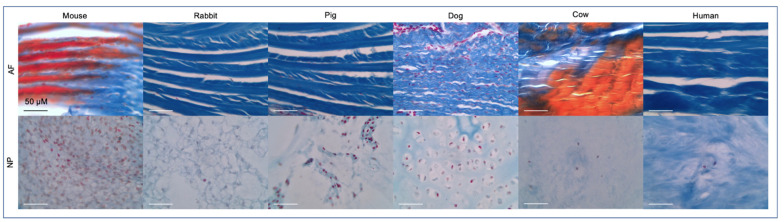
Mallory’s tetrachrome histological staining on formaldehyde preserved tissues was performed as previously described [14] to visualize the AF and NP tissue across species. All but the human sample were coccygeal IVDs. Ages were generally from adult organisms but varied across species. The dog IVD was from a 1-day old non-chondrodystrophic boxer docked breed. Scale bar reflects 50 μm. Staining represents: nuclei (red), collagen fibrils, ground substance, cartilage, mucin and amyloid (blues), erythrocytes and myelin (yellow) or elastic fibrils (pale pink, pale yellow or unstained) [54].

### 2.5. Multi Species scRNA seq Supports Cell Heterogeneity and Clarifies Cell Identities in the IVD

The physical and molecular composition of the IVD and its largely avascular and non-innervated nature creates a harsh cellular environment which impedes regenerative capabilities of the IVD. Identifying differences in biochemical and structural contributions to the NP, AF, and CEP is important for understanding how genes and proteins contribute to cell type specific signaling and maintenance of the tissue, as well as how their differential expression can support or resist degeneration [2]. While there are some conserved markers across species, a number of proteins and biomarkers are differentially expressed, and it is therefore beneficial to document markers in a species-specific manner. A major challenge in identifying biomarkers of the IVD is that the heterogenous cell population leads to subclusters of NP, AF, or CEP cells with differential expression patterns. Recent advances in scRNA seq enabled the identification of IVD cell subpopulations, including NP stem-like progenitor cells, AF progenitor cells, CC-like NP cells, endothelial cells (EC) in the oAF, myeloid cells, and lymphoid cells, in both healthy and diseased discs of various species [35,39,55,56,57,58]. Recent scRNA seq analysis makes use of statistical analysis such as T-distributed stochastic neighborhood embedding (tSNE) to characterize clusters or subclusters of NP, AF, and CEP cells based on their expression profile [39,47,59,60], however labeling and defining these cell groupings currently varies between research groups and experiments. Nonetheless, scRNA seq is a powerful technology that effectively characterizes cellular changes via gene expression changes and shifts in ratios of cell populations during degeneration helping to clarify cell identities and monitor senescence during IVD development and degeneration.

Prior to scRNA seq, Sakai et al. utilized colony forming assays to identify NP stem-like and progenitor cells in mouse and human via TEK receptor tyrosine 2 kinase (Tie2/TEK) and disialoganglioside 2 (GD2) as markers [55]. They additionally identified angiopoietin-1 (Ang1) as an important Tie2 ligand in the bovine IVD which previously displayed anti-apoptotic effects in human NP cells [55]. The frequency of Tie2 positive NP cells decreased as disc degeneration increased, indicating that these cells are significantly impacted by IVDD and aging [55]. Another interesting advancement of this study is the ability to sort and characterize the proliferation capacity of marker positive (+) or negative (−) NP cell subpopulations (Table 1) [55]. As a result, Tie2+/Gd2−/CD24− progenitor cells were described as dormant stem cells in the mouse, human and bovine NP, with Tie2+/Gd2+/CD24− cells exhibiting self-renewal potential and stem cell properties and Tie2−/Gd2−/CD24+ cells being committed to a mature NP phenotype [32,55,61,62]. Overall, the identification of Tie2+/Gd2+ NP cells is important as a potential therapeutic target and for characterizing the IVD cell population [55].

*Tie1* and *Tie2*, which are expressed during embryonic angiogenesis, were recorded in the bovine AF by scRNA seq, marking the first time that *Tie2* was described outside of NP cell progenitors [57]. This is particularly interesting as *Tie2* is expressed in hematopoietic cells [63]. Furthermore, scRNA seq carried out for cells derived from rat AF and NP tissue identified matrix metalloproteinases MMP3, and MMP13, along with interleukin 11 (IL11) as highly expressed in AF cells, however because they are rather common in related degenerative diseases they cannot be considered exclusively as an AF biomarker [58]. Aiming to identify new cell-specific biomarkers in the oAF, iAF, and NP, gene ontology analysis indicated that rat AF cell function is mainly related to fibrosis and stress response while NP cells are related to degenerative diseases and ECM maintenance [58].

More recently, urotensin 2 receptor (UTS2R) + cells, a proposed novel progenitor NP (proNP) cell cluster marker, were located in the peripheral NP tissue which showed an enrichment for progenitor cell markers and stemness genes. These cells were largely Tie2+ or Tie2+/GD2+, indicating that most UTS2R+ cells may be considered proNP cells [39]. These initial tests were performed in mice, however the group followed up with rat UTS2R+ proNPs finding that they primarily formed fibrous colony forming units (CFU) in vitro, and in matrigel had more of a spherical formation [39]. Human UTS2R+ NP cells cultured in matrigel also had improved spherical formation compared to UTS2R− NP cells [39]. Similarly, to the recorded decrease in Tie2+/GD2+ cells in degenerated IVDs, UTS2R+ cells also showed a decrease in cell number in Grade V degenerated IVDs compared to Grade II IVDs, indicating exhaustion of the progenitors [39]. Notably administration of UTS2R+ but not UTS2R− NP cells to mouse IVD puncture models showed significant attenuation of IVDD, suggesting an important role of proNP cells, which were located in a tenascin-c (TNC) enriched ECM niche in the peripheral NP [39]. Gao et al. further supported the heterogeneity of the NP cell population in mice with their scRNA analysis defining four NP cell clusters including cluster 1: NP progenitors, cluster 2: transient NP, cluster 3: regulatory NP, and cluster 4: homeostatic NP with clusters 3 and 4 making up making up 77.2% of NP cells [39]. Cluster 3 is thought to play the biggest role in degeneration and the onset of inflammatory cascades based on its enrichment of genes implicated in angiogenesis, TNFα production, and axon guidance, however ciliary neurotrophic factor receptor (CNTFR) marked cluster 3 cells were significantly reduced in Grade V degenerated discs compared to Grade II [39].

ScRNA seq of cynomolgus monkeys and immunohistochemical (IHC) analysis of monkey and rat coccygeal IVDs enabled the expression analysis of CC-like and NC-like NP cells. Similar expression of a nuclear mediator in the sonic hedgehog (SHH) pathway, glioma-associated oncogene homolog 1 (GLI1), was shared in both CC-like and articular CC cells as well as the SHH signaling mediator, smoothened (SMO), suggesting the activation of hedgehog pathways in both articular CC and a subset of NP cells [56]. Furthermore, IHC showed expression of SHH but not indian hedgehog (IHH) in vacuolated NC-like NP cells while both SHH and IHH were detected in bone marrow (BM) cells of a cynomolgus monkey femur, suggesting BM cells as a supply source for hedgehog signaling ligands activating articular CC hedgehog signaling [56]. Notably, SHH was not expressed in all NC cells, however it is thought that NC cells may begin as SHH+ NC cells and lose SHH expression over time as SHH− cells were found in individuals older than ten years old [38,56]. Hedgehog signaling may activate the hypoxia inducible factor 1 alpha (HIF1α) pathway in both CC-like and articular CC, based on the demonstrated increase in HIF1α protein levels with the addition of SHH to primary chondrocytes [56]. Furthermore, transforming growth factor beta (TGFβ) is believed to promote expression of SHH, providing a potential link between increased TGFβ, SHH, and HIF1α in NC cells. Notably, this scRNA data came only from male monkeys.

Lin et al. analyzed changes in protein C receptor (PROCR)+ progenitor cells for CC differentiation between healthy and degenerated goat IVDs [35]. Progenitor cells were found in the oAF with three main differentiation fates of regulatory CCs, fibroblast CCs and stress CCs [35]. Notably, with degeneration, differentiation towards regulatory CCs diminished significantly, instead favoring differentiation into a cluster referred to as stress CCs [35]. Regulatory CCs activity revolved around protein synthesis, mitophagy, and promotion of TGFβ and Hippo signaling pathways in relation to cartilage formation while stress CCs showed enrichment of pro-inflammatory tumor necrosis factor (TNF) and IL17 as well HIF1 signaling pathways showing a relation between stress CCs and both inflammation and apoptosis with degeneration [35]. The trend towards stress CC in differentiation of PROCR+ progenitor cells in IVDD likely contributes to the increasingly pro-inflammatory environment of IVDD pathology [35].

Caveolin (CAV) + endothelial cells (EC) were identified in the iAF and oAF of degenerated IVDs [35]. Lin et al. described the contribution of CALCR and VEGF, among other signaling pathways, to communication between chondrocytes and CAV+ ECs. Chondrocytic secretion of SEMA3C to inactivate ECs and EC secretion of TNFSF10 to disrupt CC function indicates regulatory networks between CC and ECs in the IVD which may affect vessel infiltration [35].

**Table 1 vetsci-10-00540-t001:** Reported biomarkers and differentially expressed genes for IVD tissues. Abbreviations: ELISA: enzyme linked immunosorbent assay, IHC: immunohistochemistry, (q) RT-PCR: (quantitative) reverse transcription polymerase chain reaction, RISH: RNA in situ hybridization. Gene abbreviations and reference numbers will be provided in Appendix A.

Species	NC	NP	Inner AF	Outer AF	Method	Source
Human	-	C2ORF40, MGP, MSMP, CHI3L1, LGALS1, ID1, ID3 and TMED	MT1F, PLA2GA, EPYC, PRELP, C10ORF10, FGFBP2 and CHI3L1	-	Human single-cell transcriptome analysis	[64]
	-	-	-	COL5A1	spatiotemporal and single cell transcriptomic analysis	[29]
	SOX10, CTSK	-	-	-	Mouse spatiotemporal transcriptional analysis, human embryonic single cell transcriptomic atlas, immunohistochemistry of mouse IVD	[50]
	CD24, STMN2, RTN1, PRPH, CXCL-12, IGF-1, MAP1B, ISL1, CLDN1, and THBS2.	-	-	-	Microarray analysis and qPCR validation of human embryonic, fetal, and adult spine	[36]
Rat	-	Krrt7, Prrg4, Akap12, Cxcl3, Rab38	Mmp3, Bpifb1, Bpifa1a, Mmp13, Il11, Inhba	Fibin, Igfbp5, Tnmd, Myoc, Cilp, Lum	Single-cell transcriptome analysis	[58]
	-	CD24, Basp1, Ncdn	CD90		cDNA microarray, RT-PCR, IHC	[65]
Mouse	-	Anxa3, Cdh2, Cd44, Ca3, Slc2a1, Krt8, Krt19, CD109, and CD81, of which Cdh2, CD44, Slc2a1, and CD81 are exclusive in the NP	Tnmd, Col5a1, Col5a2, Col12a1, and Lect1 of which Tnmd, Col5a2, and Col12a1 are exclusive in the AF		Proteomic analysis of mouse lumbar and tail IVD and comparison to human IVD	[66]
	-	*Gli1*, *Gli3*, *Noto*, *Scx*, *Ptpr*, *Sox2*, *Zscan10*, *Loc101904175*	*-*	Lam1, Thy1	RISH	[67]
	*Krt8*, *Atp6v1g3*, *C1qtnf3*, *Cd55*, *Spp1*	*Cp*, *S100b*, *H2ac18*, *Snorc*, *Creld2*, *Pdia4*, *Dnajc3*, *Chcd7*, *Rcn2*	*Mgp*, *Comp*, *Spp1*, *Gsn*, *Sod2*, *Dcn*, *Fn1*, *Timp3*, *Wdr73*	*Igfbp6*, *Ctsk*, *Lgals1*, *Ccn3*	Sc RNA seq	[68]
Bovine	-	*T*, *Cd24 and Krt19*	*-*	*Adamts17*, *Col5a1*, *Col12a1 and Sfrp2*	qPCR	[69]
	*T*	*Cdh2*, *Krt8*, *Krt18*, *SNAP25*, *Sostdc1*, *Ibsp*	*-*	*-*	qRT-PCR and Microarray Analysis	[70]
	*-*	*Gli1*, *Gli3*, *Noto*, *Ptprc*, *Scx*, *Sox2 and Zscan10*	*-*	*Lam1*, *Thy1*	Fluorescent RISH, confocal microscopy, gaussian mixture modeling	[28]

## 3. Stress and Inflammation in the IVD

### 3.1. Environmental Stress Factors

The composition of an IVD’s ECM enables the disc to carry the mechanical load and flexibility required for movement, however multiple environmental and metabolic factors can impact ECM composition and remodeling. As such, maintaining proper ECM ratios and balance is necessary for preserving the healthy microenvironment and disruption of these levels relates to degeneration. Here we analyze mechanical, environmental, and metabolic stressors which alter IVD ECM and examine potential contributions to degeneration.

#### 3.1.1. Mechanical Stress and Trauma

Repetitive stress and loads on the spine are impacted by posture and movement patterns, such as bipedal versus quadrupedal movement. As bipedal vertebrates, both male and female humans experience the most stress and therefore highest prevalence of degeneration, in the lumbar region [66,71]. Hyperlordotic spines, which display an increase curvature of the lumbar spine, experienced significantly greater stress on the IVD compared to healthy models, while hypolordotic spines, which have a decrease in lumbar spinal curvature, generally experienced diminished IVD stress and minimal compression of the disc [72]. Additionally, the IVD experiences varying levels of loading stress, cycling between periods with decreased loading, typically at night, as the spine maintains a decompressed horizontal laying position and greater loading periods while assuming an upright position that leads to a decrease in overall disc height throughout the day [73]. The average combined diurnal loss of disc height due to compressive loading throughout the day was 5.91 mm [73]. Changes in spinal loading are important as they enable the cyclical expulsion and intake of fluid in the disc which affects solute diffusion and transportation of nutrients to the different disc tissues [16]. However, efficiency of solute diffusion is impacted by degeneration. IVDs demonstrated a significant decrease in solute diffusion with calcification of the CEP, showing an inverse relationship between solute diffusion and CEP calcification levels [74]. Thus, illustrating that physical loading changes in the disc can impact solute diffusion and nutrient transport [10].

When considering compressive loading of the spine, it is important to also consider how the stress is dispersed throughout a tissue. For example, uneven loading of a bovine IVD, producing concave and convex sides, primarily stressed the AF in the concave side, leading to decreased AF tissue, increased caspase-3 (CASP3), and reduced ACAN, while in the convex side, MMP1, ADAMTS4, IL1β, and IL6 mRNA was upregulated in the AF but not the NP compared to controls [75]. This indicates not only a tissue specific response to loading forces but also demonstrates how uneven stress can influence the expression of pro-inflammatory cytokines. Furthermore, bovine coccygeal IVDs cultured in TNFα containing medium and exposed to dynamic loading resulted in a significantly increased percentage of TNFα+ cells in the NP compared to control or static loading groups, indicating the importance of convective transport for TNFα penetration into a healthy NP [76]. Notably, increased TNFα levels in NP cells promoted ACAN degradation and a significant increase in IL6, IL1ß and TNFα production as a sustained inflammatory effect of TNFα [76]. Spring loaded compressive force applied to canine spines for up to one year did not produce changes in PG or collagen compared to controls [77]. However, static compressive forces applied to rat tail IVDs resulted in p53 mediated apoptosis and decreased NP and AF cell numbers as well as a decreased proportion of cells expressing NC markers [15]. This further suggests that the type of loading greatly impacts the IVD’s stress response.

Beyond compressive forces, Rohanifur et al., 2022 described cellular remodeling after lumbar disc puncture (LDP) induced trauma through changes in gene expression. Sequencing compared pooled male rat lumbar (L) 4-L5 IVDs following LDP to L2-L3 unpunctured control discs [78]. A shift in cell clusters was noted eight weeks post LDP with newly identified ECs and an increase in the main cluster of IVD cells in the treatment group at the expense of cells contributing to the myeloid and lymphoid clusters. Interestingly, subclusters of AF cells expressing nerve growth factor (Ngf) and its receptor (Ngfr) increased in degenerated discs. The authors speculated that the increased number of cells expressing Ngf and Ngfr, in combination with the newly registered lymphoid cells in degeneration, may suggest signaling between neuronal cell population and immune cell population during degeneration and injury to the disc [78]. Beyond age-based changes, Moseley et al. sought to determine sex-based associations between annular puncture model of IVDD and pain. Male and female rats were subjected to annular disc puncture or sham surgery to determine sex-association of IVDD induction, radiologic IVD height, histological grading, or biomechanical testing [79]. Female rats showed the greatest association between injury, histological grading and IVD height, while male rats additionally showed significant association with von Frey thresholds, linking injury to pain levels. Furthermore, evidence suggested the decrease in IVD height is stronger in more caudal discs, indicating significant disc level variation that should be accounted for when creating experimental models [79]. Lastly, considerable variation in methods of inducing IVDD in animals has been shown to produce differing IVDD phenotypes [80], which may result in altered gene expression profiles and biomarkers for natural occurring IVDD. The use of marker gene panels for each tissue type is therefore recommended. Overall, this highlights the importance of considering both age and sex of animals when determining experimental methods of inducing IVDD, including type and distribution of loading stresses, the size of instruments used for disc puncture and the level of the IVD examined.

#### 3.1.2. Metabolic Stress

The avascular NP establishes a harsh environment with cells depending on anaerobic lactic acid fermentation that promotes a decrease in pH [48,81]. However, a decrease in lactic acid production in isolated bovine NP cells following acidification of the medium or low oxygen content was described, indicating a negative Pasteur effect [82]. Furthermore, oxygen consumption rate in bovine NP cells in medium is affected by both oxygen content and pH, with oxygen consumption significantly decreased in environments with lower pH and lower oxygen [82]. Typically, acidification of the IVD accompanies degeneration; in human IVDs, the pH of a healthy disc is approximately 7.1 while degenerated discs have a pH of 6.5 and severely degenerated discs were recorded with a pH as low as 5.7 [83]. Matrix synthesis disruption occurs with disc acidification as demonstrated by the significant drop in PG synthesis when the pH falls below 6.8 [81]. Overall, the decreased PG content with disc acidification can further contribute to degeneration through the related decrease in IVD osmotic pressure that disrupts load bearing and solute transportation abilities [81]. Acidification of the IVD further leads to increased production of known pro-inflammatory markers with an 81-fold increase in IL1ß, 7.8-fold increase in IL6, 3-fold increase in NGF and 4.6-fold increase in BDNF when cells were cultured at pH 6.5 compared to controls [83]. At pH 6.2, bovine NP cells showed a slight reduction in viability, however human NP cells experienced cell death at both pH 6.5 and 6.2, no proliferation at pH 6.8 and effective proliferation at pH 7.1 and 7.4 [83]. This suggests that an acidic pH has a strong effect on promoting expression of matrix degrading proteins, pro-inflammatory cytokines, promotes cell death, and prevents NP proliferation in humans. Furthermore, it indicates a species dependent difference in the pH levels that trigger cell death and degenerative changes, which should be clarified in other species. Fermentation provides less usable chemical energy in form of adenosine triphosphate (ATP). This crucial energy providing nucleotide influences molecular levels of PGs, collagen, and other ECM molecules [84]. As previously mentioned, disruption of these levels, such as decreased PG, can contribute to degeneration. Bovine IVD cells cultured with 100 µM ATP for two hours showed significant increases in internal ATP content in NP and AF cells, with NP cells showing higher levels than AF cells in both control and experimental groups [84]. This suggests extracellular ATP treatment leads to intracellular ATP increases. Interestingly, 5x more ATP was needed in NP versus AF cells to show significantly increased PG and collagen levels compared to controls [84]. Higher mitochondrial protein activity in the mouse NP suggests potentially higher mitochondrial activity in the NP region at large, thus offering a potential explanation for the generally higher ATP concentration required for influence on ECM synthesis [66]. While reduced mitochondria were described in human notochord cells [30] fewer mitochondria in IVD derived compared to adipose tissue derived cells were noted but not quantified (Figure 4). Furthermore, 16 h of ATP treatment led to significant increases in Acan and collagen II gene expression in both AF and NP cells [84]. Overall, this indicates that extracellular and intracellular levels of ATP influence levels of PG, collagen, and Acan, thus altering ECM composition.

### 3.2. Inflammation and Degeneration

#### 3.2.1. Regulated Cell Death and Tissue Homeostasis

Regulated cell death is an important process to maintain homeostasis of any tissue under stress [7,8]. Increased cell death is common in IVDD, of which inflammasome-mediated, caspase-dependent pyroptosis is thought to promote a pro-inflammatory response [85]. In pyroptosis, nucleotide-binding domain and leucine-rich repeat (NLR) and nod-like receptor protein 3 (NLRP3) recruit caspase-1 (Casp1) [85]. This promotes pore formation in the plasma membrane, allowing for a quick water influx, creating swelling and subsequent cell lysis [85]. Pyroptosis is associated with the release of activated IL1β and IL18 following cell lysis and is further linked to matrix degradation, NP cell apoptosis, CEP tearing, vascularization, and nerve ingrowth [85]. Milk-fat globule-EGF factor 8 (MFG-E8), a modulator of the NLRP3 inflammasome, inhibits pyroptosis in NP cells and IVDD [86]. Furthermore, a positive correlation exists between disc cell apoptosis and caspase-12 (Casp12), GRP78 expression, and cytochrome C mitochondrial release in rats [87]. The induction of AF cell apoptosis via sodium nitroprusside resulted in upregulation of Casp12, GRP78, and GADD153 while cytochrome C levels increased in the cytosol [87]. Ferroptosis, cell death through lipid peroxidation initiated by iron, was first described in 2003 but not coined until 2012 [8,88,89]. Understanding the specific role of ferroptosis in IVDD onset and development is ongoing, however studies using animal models support a role for ferroptosis in IVDD. AF and NP cells in a simulated oxidative stress environment demonstrated an upregulation of prostaglandin endoperoxidase synthase 2 (PTGS2) and downregulated glutathione peroxidase 4 (GPX4) and ferritin heavy chain (FTH), suggesting ferroptosis in the degenerated IVD [90]. Additionally, CEP degeneration and IVDD in mouse iron overload models showed degeneration occurred in a dose dependent manner [91]. In addition to pyroptosis, apoptosis, and ferroptosis, necroptosis also contributes to regulated cell death in IVDD. Necroptosis occurs due to death receptor activation including TNFα activated tumor necrosis receptor (TNFR1) and Fas, or Toll-like receptors TLR3 or TLR4 [7]. Necroptosis markers were found in human degenerated discs [92]. MyD88 was found colocalized with necroptosis markers and inhibition of MyD88 rescued necroptotic NP cells, suggesting a role of MyD88 in IVDD [92]. Unlike ferroptosis, necroptosis activation is not a mitochondrial influenced process [7], however necroptosis can impact mitochondrial ultrastructure [92]. Regulated cell death including pyroptosis, ferroptosis, necroptosis and apoptosis in the IVD promote pro-inflammatory cytokines and activation of caspases, responsible for signaling and inflammatory cascades, autoimmune responses, and cell death [7,85,87,93] which further accelerates degeneration of the IVD.

#### 3.2.2. Protagonists and Antagonists in the Proinflammatory IVD

Damage-associated molecular patterns (DAMPs) are a group of trauma or tissue damage response molecules, including ECM fragments recognized by pattern recognition receptors (PRRs) that activate signaling pathways and trigger the release of inflammation mediators [94,95]. As such DAMPs activate microglia and macrophages in aging and chronic inflammation and exasperate inflammation through pro-inflammatory cytokine release via NLRP3 and TLRs [96,97,98] (Table 2). A 2020 study used isolated NP cells from canine herniated IVDs and exposed them to 30 kDa fibronectin fragments (FnF) which stimulated a significant upregulation of IL6 and IL8; while the response varied between donors this suggested FnF contributed to a pro-inflammatory environment and may offer another contributing family of molecules that assist in degeneration [98].

Phagocytotic macrophages have complex contributions to inflammation and IVDD. In response to injury, macrophages transition from a pro-inflammatory M1 phase to an anti-inflammatory M2 phase, further categorized as either M2a or M2c, of which M2c is thought to assist with tissue healing, yet secreting MMP7, MMP8, and MMP9, known for pro-inflammatory and ECM remodeling effects [99]. Inflammatory response to an increase in expression of pro-inflammatory cytokines is a major contributor to IVDD. Pro-inflammatory cytokines such as TNFα, IL1β and IL6 amongst others are predominantly produced by activated macrophages [99]. Mediating this response is one method for mitigating inflammation and IVDD.

Cells positive for CCR7, a cell surface marker that is associated with pro-inflammatory M1 and CD206+ M2a cells both considered to have anti-inflammatory properties, did not significantly increase with degeneration [99], hence suggesting that IVDs fail to reach the associated anti-inflammatory and tissue repair stage. CD163 associated with M2c, were significantly increased in degenerated NP, AF, and CEP tissues compared to healthy tissues according to the severity of degeneration and may be beneficial markers for degeneration and inflammation [99]. Interestingly, both M2a and M2c macrophages are associated with tissue repair. However, they are also associated with various diseases including spondyloarthropathy and pulmonary fibrosis among others and their specific impact remains an area of active research. Macrophages might enter through degenerated CEP [99].

To better understand the impact of macrophages on the inflammation of bovine IVDs, Silva et al. cocultured bovine IVD biopsies in IL1β containing medium which led to increased IL6, IL8, and MMP3 levels, however when cultured in IL1β containing medium with macrophages an insignificant reduction in IL6 and IL8 expression was noted and no increase in MMP3, suggesting macrophages interfered with the production of MMP3 and tissue remodeling in pro-inflammatory environments [100]. A different study cultured human NP cells and mouse IVDs in either fetal bovine serum (FBS), TNFα, or TNFα and M2 macrophage conditioned medium. The co-cultured group had an upregulation of ACAN, and type II collagen compared to TNFα only and indicated a positive effect of M2 macrophage conditioned medium on ECM synthesis [101]. TNFα upregulated MMP13, ADAMTS4, ADAMTS5, and IL6 while the TNFα and M2 macrophage co-culture groups reversed the pro-inflammatory effect [101]. These studies show a potential for macrophages to alter inflammatory responses in the IVD.

Injection of TNFα into bovine IVDs led to upregulation of MMP3, COX2, IL6, IL8 and ADAMTS4 [102]. Interestingly, a significant increase in COX2+ cells was reported in severely degenerated IVDs compared to healthy or mild to moderately degenerated IVDs in humans and dogs [103,104], supporting it as a marker of severe degeneration. Isolated bovine NP cells treated with either TNFα or the inflammatory agent lipopolysaccharide (LPS) and subjected to decreases in osmotic loading produced sustained changes in F-actin expression in treated cells compared to controls [105].

The monomeric form of C-reactive protein (mCRP) is commonly used as an acute phase marker for progression of inflammatory diseases due to its strong proinflammatory nature in chondrocytes [106,107]. Compared to the native pentameric CRP, both the intermediate pentameric CRP (pCRP*) and mCRP showed pro-inflammatory effects, especially in CCs [106,108]. In recent years, there has been a rise in studies exploring the prevalence of CRP in inflammatory mediated degenerative conditions. A study on patients recommended to undergo lumbar spinal surgery due to single level discopathy had a higher prevalence of blood CRP levels compared to those not recommended while excluding individuals with multiple levels of discopathy or rheumatologic conditions from the study [109]. CRP levels, if tied to inflammatory progression, would likely be elevated in individuals with multiple affected discs or higher Pfirrmann Grades, similar to mCRP levels being associated with the severity of OA. Degeneration of articular cartilage is a feature of both OA and IVDD [107]. Ruiz-Fernandez et al. were the first to functionally confirmed the presence of CRP in both human AF and NP cells [106]. Furthermore, nitric oxide, known for its upregulation of MMPs and cytotoxicity and promotion of ECM degradation, is increased with activation of mCRP in addition to MMP13, IL6, IL8, and LCN2, demonstrating a correlation between mCRP and inflammatory mediators associated with increasing degeneration [106]. Pathways important for mCRP signaling in the AF include NF-kB, ERK1, ERK2, and PI3K [106]. To understand the potential role of CRP in other animals, follow up studies will be necessary. CRP may be a useful acute marker of inflammation associated with IVDD; however, it should be noted that CRP plasma levels are also increased in rheumatoid arthritis, infection, cancer, and tissue trauma [110]. Lastly, a recent study designed C10M, a low molecular weight compound, to block pCRP binding to the phosphocholine groups of damaged cell membranes that mediate the conformational change between pCRP to pro-inflammatory pCRP* and mCRP [108]. Thus, presenting a possible anti-inflammatory tool to combat CRP induced inflammation in IVDD.

While many cytokines have been shown to promote inflammation, not all cytokines promote IVDD. Utilizing a rat caudal IVD puncture model to determine the effect of the anti-inflammatory chemokine IL10 on IVD degeneration, rat IVDs were injected with 20 microliters of IL10, or a saline solution as a control on a weekly basis [111]. Among sections of IVDs from degenerate, IL10, and saline treated groups, IL10 groups showed decreased levels of p38 mitogen activated protein kinase (MAPK) and collagen X in addition to higher collagen II expression levels [111]. This finding was further supported by the maintenance of a healthy ring-like AF structure in IL10 treated cohorts [111]. Findings from MRI, histological and IHC analysis all suggested a positive effect of IL10 in promoting a healthy IVD microenvironment and indicating reversal of disc degeneration [111]. IL1ß induced NP degeneration increased transcription of *Col10* and decreased levels of *Acan* and *Col2* mRNA levels [111]. Samples treated with both IL1ß and IL10 to determine potential anti-inflammatory effects of IL10 did not alter *Acan* mRNA levels significantly, however it increased *Sox9* and *Col2* expression while decreasing *Col10* levels [111]. Following an initial stress response, injection of IL10 treated samples showed a significant decrease in p38 phosphorylation as well as lowered p38 MAPK activation [111]. This suggests anti-inflammatory effects of IL10 in rat IVD through a reduction in p38 stress activated signaling. Interestingly, in humans, IL10 was found to increase sensitization of the IVD [112,113], indicating deviation between species in response to IL10 treatment.

Evidently, dynamic loading, biomechanical and metabolic stress affects IVD tissue through nutrient diffusion, decreased ECM protein synthesis, increased pro-inflammatory cytokines and apoptosis and diminishing cell numbers, leading to long-term changes of ECM organization that can alter disc height, hydration and movement of solutes. Hence resulting in sustained physical and biochemical changes of the IVD promoting IVDD across species. Continuing to understand how these individual facets of stress function together to promote IVDD pathophysiology ultimately is important for determining a functional therapy.

**Table 2 vetsci-10-00540-t002:** This table describes several differentially expressed genes and proposed markers for degeneration and inflammatory responses in IVDD compared to healthy or herniated discs as well as the species in which they were determined. Abbreviations: ELISA: enzyme linked immunosorbent assay, IHC: immunohistochemistry, (q) RT-PCR: (quantitative) reverse transcription polymerase chain reaction, RISH: RNA in situ hybridization. Gene abbreviations and reference numbers will be provided in Appendix A.

Species	Degenerating IVD	Method	Source
Humans	IL6 serum level is significantly higher In IVDD than disc herniation or control groups	Electrochemiluminescence immunoassays	[114]
	Elevated CCL5 and CXCL6 plasma levels in moderate to severe IVDD compared to healthy control	ELISA and MRI	[115]
	Significant increase in IL18 for degenerated Grade IV/V IVD	IL18 ELISA kit	[116]
	Degeneration: MT1G, SPP1, HMGA1, FN1, FBXO2, SPARC, VIM, CTGF, MGST1, TAF1D, CAPS, SPTSSB, S100A1, CHI3L2, PLA2G2A, TNRSF11B, FGFBP2, MGP, SLPI, DCN, MT-ND2, MTCYB, ADIRF, FRZB, CLEC3A, UPP1, S100A2, PRG4, COL2A1, SOD2 and MT2A Verified by protein and mRNA expression: MGST1, vimentin, SOD2 and SYF2	scRNA seq, quantitative immunofluorescence and Western blotting	[64]
	Shear stiffness in both NP and AF correlated with increased Pfirrmann Grade of degeneration	MR elastography shear stiffness measurement	[117]
	Cox2+ cell number from Grade 2 IVDD onwards	IHC	[104]
	Reduced *GRB10* in lumbar IVDD compared to healthy controls. Not detected in piriformis syndrome, sacroiliac joint pain, entrapment neuropathy and lumbar disc herniation, suggesting a biomarker for lumbar IVDD.	miRNA qRT-PCR from plasma sample	[118]
	CASP1, IL1β, NLRP3	mRNA expression, IHC	[119]
	SPP1 secreted by NC	Mouse spatiotemporal transcriptional analysis, human embryonic single cell transcriptomic atlas, IHC of mouse IVD	[50]
	Increased sensitization of the disc: IL1β, IL6, IL8, IL10, TNFα, IFNy Upregulated in Disc herniation group compared to IVDD: IL4, IL6, IL12, IFNy	IHC	[112,113]
	Increased levels in IVDD compared to herniated NP (HNP) groups: TNF-α and IL8 Increased expression levels in IVDD compared to herniated NP (HNP) groups: TGFb, VEGF, and NGF	Western Blot	[120]
	CCR7+ and CD163+ cells significantly increase with degeneration in NP, AF, and CEP, while CD206+ cells were present but did not significantly increase with further degeneration. CCR7+, CD163+, and CD206+ cells were not found in healthy IVD.	IHC	[99]
	Upregulated in degeneration: SLC7A2, LIF, NAMPT, IL1β, NOD2, CCL20, CCL7, TNFRSF1B, LYN, and GCH1 Inflammatory genes proposed as biomarker of IVDD with positive correlation to infiltration of immune cells: IL1β, LYN, NAMPT	Statistical analysis of available gene expression profiles	[121]
	Decreased expression in degenerate IVD: *CDH2*, *KRT8*, *KRT* Increased expression in degenerated AF: *VCAN*, *TNMD*, *and BASP1*	qRT-PCR and Microarray analysis	[70]
Humans and Rats	Upregulated in dNP: COMP, MGP, FBLN1, BASP1, NCDN and CD155 Downregulated in dNP: SNAP25, KRT8, KRT18, CDH2, KRT19, NRP-1 and CD221	Cumulative data from rat RT-PCR, human RT-PCR, and rat microarray	[122]
Rat	Core genes in IVDD in TSZ-induced T1DM rats: Bmp7, Ripk4, Wnt4, Timp1, Col11a1, Acp5, Vdr, Col8a1, Aldh1a1, and Thbs4	Microarray analysis and transcriptome sequencing of NP cells followed by interaction analysis	[123]
Dog	*Clusterin* as a cerebrospinal fluid marker for chronic IVDD	Liquid chromatography, mass spectrometry, SDS-page, Western blot, and IHC	[124]
	Significantly lower GAG content in Grade IV and V degenerated discs	GAG assay	[125]
	Cox2+ cell percentage in the NP and dorsal AF is significantly higher in Grade IV and V discs compared to Grades I and II	IHC	[125]
	Significantly higher protein levels of IL8 and Tnfα in IVDD versus healthy discs. Significantly higher mRNA levels of *IL6* and positive correlation between IL6 and pain severity	qPCR, ELISA, IHC	[103]
	Increased levels of Tnfα in discs adjacent to IVDD or herniation	qPCR, ELISA, IHC	[103]
Cow	Lam1, Thy1	RISH	[67]
	Significantly lower number of CD29+, CD44+, CD45+ and Tie2+ aged NP cells compared to younger NP cells	Flow cytometry, IHC	[126]

## 4. Insights from Different Species

While IVD tissues of different species appears relatively consistent, with most dissimilarities found in the NP (Figure 2 and Figure 3) the variation in movement patterns, localization of highest IVDD prevalence, response to interleukins, and NC-like cell disappearance between species indicates the importance of considering IVDD research advancements in the context of species-specific research as well as the consideration of research methodology for induced degeneration. The following section will discuss structural and molecular similarities and differences (Table 2) as well as developments in IVDD research in a species dependent manner. It is notable that the extent of research varies by species with some having only recorded the first cases of IVDD in the last decade.

### 4.1. Primates

#### 4.1.1. Humans

While spine development is typically considered to occur in a rostro-caudal direction, a 2018 study proposed a verifiable newly described process for NC development in humans (*Homo sapiens*) [127]. Starting with formation of the NC process (day 17–23) its thickening and an epithelial to mesenchymal transformation (EMT) marks the prechordal plate stage [127]. Once embedded into the endoderm, the NC process is considered the NC plate (day 19–26) before it further develops into the definitive NC proper (day 23–30) [127]. The NC remains in physical contact with the ectoderm derived central nervous system (CNS) throughout NC development as it attaches to the “ventral floor” of the CNS, [127]. The NC plate, NC process, and somites are thought to develop from the cranial end and progress caudally [127,128]. However, de Bree et al. proposed a bidirectional developmental shift away from craniocaudal development by 23–30 days, instead favoring a central origin of development that progresses in both the cranial and caudal directions [127]. This allowed the NC ridges to fully close in the central region of the embryo forming the definitive NC fully released from the endoderm and cranially released from the neural tube by day 26 of development [127]. The secretory activity of NC cells begins around day 34 and peaks by 50 days in humans, enabling the production of a NC sheath [129]. Overall, this implicates all three germ layers in NC development, offering a potential explanation of the complex molecular nature of IVD cells.

Human IVDs are subject to naturally occurring degeneration, producing pain in a large portion of the population. As in other species, the pathology of IVDD is still under research. Differentially expressed genes in IVDD patients, such as lowered GRB10, increased serum levels of markers such as IL6 and ferritin, and other notable differences can contribute to IVDD diagnosis (Table 2) [91,114,118]. Notably, all human studies in Table 2 examined cases of naturally occurring IVDD in humans. As the complexity of IVD tissue and IVDD degeneration has become apparent, it is important to consider how induced degeneration, such as through LDP, may activate only a portion of the degenerative processes that contribute to IVDD. Therefore, naturally occurring degeneration may show different biochemical and or clinical presentations from induced IVDD.

#### 4.1.2. Non-Human Primates

Rhesus monkeys (Macaca mulatta) often used in research have seven cervical vertebrae, 12 thoracic, seven lumbar, three fused vertebral bodies in the sacral spine, and approximately 20 caudal vertebrae [130]. Naturally occurring IVDD has been recorded in rhesus monkeys [131]. Furthermore, naturally occurring degeneration occurs in baboons, with severe degeneration after 14 years of age that both radiographically and histopathologically resembles the degeneration of human IVD [132]. While IVDD occurs spontaneously in rhesus and cyanomolgus monkeys, anterolateral annulus resection can also be performed to induce IVDD for research purposes [131].

Organization of the IVD is largely similar in monkeys and humans. Thicker collagen fibrils (70–110 nm) localized to the oAF and CEP while thinner fibrils (40–50 nm) localized to the iAF and the outer NP in thoracic male rhesus IVD [130]. Localization of collagen I and collagen II remained the same across disc levels (C5-C6, T3-T4, T9-T10, L2-L3, L4-L5), however disc height was significantly lower in the cervical discs, registering only 55% that of the lumbar disc [133]. This indicates significant disc differences that are important for consideration, especially in studies using different disc levels as controls compared to experimental groups. In cynomolgous monkeys, fibronectin is found in the CEP, oAF but not the iAF or NP [134]. Interestingly, radial tears in the AF associated with degenerated discs in humans, but not in rhesus monkeys which may be due to postural differences as the rhesus monkey uses quadrupedal movement, however they mimic spinal loading of humans when sitting [135]. Lastly, sex-based differences of degeneration were identified in monkeys. Male rhesus monkeys had significantly more severe osteoarthritis compared to their female counterparts, as well as a significantly higher prevalence of OA [136]. Overall, this supports non-human primates as a beneficial model to study human IVD, disc level differences, and sex-based differences in certain manifestations of degeneration which should all be considered during experimental model determination.

### 4.2. Dogs

Dogs (*Canis lupus*) have a long history of IVDD and protrusion or extrusion of the disc [125]. The first reported diagnosis of IVDD in canines, then termed echondrosis intervertebralis, was reported in 1896 by Herrmann Dexler as cited in [137]. Since it’s identification, IVDD has remained a major contributor to pain in dogs. Recent studies, report that IVDD affects approximately 2–5% of all dogs, with higher prevalence in specific breeds [138]. IVDD and degeneration in canines is typically evaluated with consideration of chondrodystrophic (cd) and non-chondrodystrophic dog breeds (ncd). Between these groups, presentation of IVDD is different both in onset and clinical findings. Structurally, chondrodystrophic breeds have extremely short limbs compared to their torso length [139]. Additionally, cd breeds typically experience NP extrusion as early as two years of age and early signs of ECM breakdown by three months, while ncd breeds more often experience AF protrusion and IVDD at five years or older [139]. NP extrusion, enabled by the calcification of NP cartilage and dehydration, in IVDD is considered Hansen Type I while AF protrusion is classified as Hansen Type II IVDD [10,138,139,140]. Radiographic identification of IVD calcification in many cases can be used in the visualization of acute disc extrusion [10], however CT is suggested as a superior diagnostic tool in dogs presenting with signs of IVDD [141]. In a study with 25 extruded dachshund discs, radiographic evidence of calcification was found in 68% while both CT and histopathology identified calcification in 100% [141]. This suggests all extruded discs in dachshunds to have calcification and supports CT as a superior diagnostic device for identification of IVDD compared to radiography [141]. Interestingly, three variants have been associated with disc calcification in Danish wire-haired dachshunds, a single nucleotide polymorphism in the 5′ untranslated region of *KCNQ5* and two in *MB21D1* [142]. Identification of variants associated with disc calcification may assist selective breeding practices to mitigate higher genetic probabilities of IVDD in some dachshunds [142]. Notably, radiographic reports of calcification most frequently are described in the thoracic spinal region, with highest frequency between T10 and T13 [143]. An additional study examining reported highest localization of IVD herniation to the thoracolumbar and lumbar spinal regions with 46 out of 60 and 14 out of 60 dogs respectively [144]. Interestingly, the canine ventral AF is 2–3 times thicker than the dorsal AF, thus disc herniations are more likely to occur on the dorsal side, allowing nerve impingement and compression of the spinal cord [6,16]. Breeds of cd and ncd dogs do appear to share some aspects of IVDD development. A 2017 study concluded that both, cd and ncd dogs experience chondroid metaplasia in the NP, as indicated by the lack of fibrocytes across degenerative stages [145]. This contradicts previous work describing NP degeneration in ncd dogs through fibrous metaplasia [137,143]. Notably, a study comparing 8117 reported cases of IVDD in canines, found Dachshunds, Pekingese, Beagle, Welsh Corgi, Lhasa Apso, and Shih Tzu breeds to have a significantly greater risk of developing IVDD [146]. Their longer spines and shorter legs could produce heightened strain on the spine if lacking muscular support, especially with quick jarring movements or excessive loading [140,147].

Functionally, the NP, AF, and CEP in canines is largely consistent with other species. When compressed through mechanical loading, the high osmotic pressure in the NP is contained by the AF and CEPs, preventing extrusion [16] and the IVD is further supported and loaded by trunk muscles that enable spinal movement including flexion, extension, and twisting [16]. Due to the lack of blood supply to the IVD, iAF and NP regions rely on CEP transmission of oxygen and glucose, while the outer vascularized portions of the AF can intake additional nutrients ([16]. Fluid flow carrying albumin and large molecules is maintained as the IVD experiences loading [16]. Similar to humans, negatively charged aggrecan attracts water into the bean shaped canine NP resulting in an osmotic gradient, with 80% water in the NP and 60% water in the AF of a healthy IVD, and approximately 50–80% water in the CEP [16]. Like humans, dogs experience a shift from NC cells to CC-like NP cells however this typically occurs towards the earlier part of a dog’s life with most cells replaced by one year, and degeneration expected far later in cd breeds [125]. Additional scRNA seq of canine IVDs in healthy and degenerative states would be helpful to further clarify the shift from NC-like to CC-like NP cells in aging and degeneration. Similar to humans, the TZ also blends the mucoid and fibrous phenotype of NP and AF, respectively [16].

### 4.3. Rodents

#### 4.3.1. Mice

Much of our understanding of IVD development comes from genetic engineering and fate mapping experiments performed in mice (*Mus musculus*) [34]. Compositionally, mice were described to have seven cervical, 13 thoracic, six lumbar, four sacral, and 28 caudal vertebrae [148,149]. Mouse IVDs, however, are significantly smaller than human discs, rendering some analysis challenging. Due to the small size of the mouse IVD, total RNA from each disc may require pooling with adjacent discs to obtain the necessary amounts of RNA for the molecular assessment of transcriptome changes and the analysis of MMP, matrix markers, and inflammatory cytokines [150]. As such, RNA in situ hybridization (RISH), IHC and immunofluorescence may be favored over quantitative reverse transcription polymerase chain reaction (qRT-PCR) for those seeking a narrower analysis or analysis at the single cell level. As the heterogenous cell population often requires single cell resolution and cell pooling may mask important differences in gene expression [14,150].

Further physiological differences exist between humans and mice. Proteomic profiling comparison between mice and humans showed some divergence in the NP region, however AF data was largely consistent [66]. The proteomic divergence of the NP may be explained through varied loss of NC cells. For many years, mice were considered to retain their NC cells for a larger proportion of their life while humans are thought to lose theirs by ten years old (Figure 2) [150]. However, when defining the disappearance of NC cells, consistent definitions of NP cells are crucial: NC cells which reside in the transient NC sheath, NC-like NP cells which have similar expression profiles to NC cells, and CC-like NP cells [3]. Mice and humans share COL12A1 as an AF marker and KRT8, KRT19, CD109 as NP markers (Table 1) [66]. Further differences exist between mouse coccygeal and lumbar IVDs which are round and kidney shaped, respectively [151]. Additionally, lumbar mouse IVDs had greater enrichment for mitochondrial proteins compared to tail discs, with the greatest differences in the NP region [66]. In coccygeal IVDs, the AF had little variation in thickness around the circumference of the disc, while in the lumbar mouse IVD the AF was thicker ventrally compared to dorsally [151]. This thicker ventral AF region in mouse lumbar IVDs is consistent with findings in canines [16]. Additionally, Brendler et al. noted chondrocytic eosinophilic type cells in the NP [151]. If eosinophilic cells are present within the mouse NP IVD, it would challenge the idea that the NP is an immune privileged tissue. While the cell type was noted in 3-month-old mature mouse in this study, the IVDs were not considered significantly degenerated [151]. As such, it may be beneficial to explore the potential presence of eosinophilic cells in the NP of younger mice. Because the disappearance of NC and NC-like NP cells is linked to the onset of IVDD in humans, further research may be beneficial to understand the effects of retaining NC cells in addition to any potential effects of NC differences between lumbar and coccygeal IVDs.

#### 4.3.2. Rats

Rats (*Rattus norvegicus*), often of the Sprague Dawley or Wistar breed, are research staples [152] and have been reported to retain NC cells throughout their lives (Figure 2) [58]. Rats are commonly used as a model organism in IVDD research. Since rats have larger IVDs than mice, they are favored for analyzing therapeutic injections [80]. Careful methodology determination for surgery, puncture models, and injections is necessary to elicit the desired IVDD phenotype. Needle size for LDP induced IVDD impacts the severity of degeneration and onset of pain hypersensitivity [80]. Use of an 18-gauge (G) needle is required to elicit severe degeneration with hypersensitivity to mechanical stress in female rats [80]. Additionally, age dependent changes in protein expression analysis, determined through cDNA microarray, qRT-PCR, IHC analysis and histological stains, support variation in NP protein expression [65]. As such age dependent IVD tissue biomarkers are helpful in proper characterization of the IVD and IVDD. A decline of neurophillin-1 (Nrp1) was detected in mature rats compared to younger controls [65,122]. NRP1 as receptor for SEMA3A, thought to perpetuate an avascular environment, plays a role in regulating MMP13 expression in human chondrocytes [153]. Furthermore, *brachyury* (*Tbxt*) mRNA, was detected in all analyzed age groups, however protein was only detected at 1 month suggesting the importance of brachyury protein in context of NC cell loss with aging [65]. *Basp1*, *Ncdn*, and *Arp1*, based on their expression in immature NP cells and their neuronal-association, were recommended as potential NC-like NP markers [65].

Diabetes mellitus has been implicated as an indicator/contributor to IVDD. To understand diabetes mellitus type 1 (T1DM) induced IVDD, Yu et al. confirmed IVDD in streptozotocin (STZ)-induced T1DM rats and performed NP transcriptome analysis in conjunction with microarray screening to identify genes relevant to IVDD in the T1DM rats [123]. This resulted in 35 potential genes of interest which were enriched in ECM, cytokine adhesion binding, and displayed prominent molecular function in apoptosis regulation and morphogenesis [123]. Interaction analysis of the 35 top differentially expressed genes (DEG) revealed *Bmp7*, *Ripk4*, *Wnt4*, *Timp1*, *Col11a1*, *Acp5*, V*dr*, *Col8a1*, *Aldh1a1*, and *Thbs4* as the ten core genes in IVDD in STZ induced T1DM rats in descending enrichment [123]. Further ELISA results found pyroptosis-related proteins NLRP3, IL18, IL1β, and gasdermin-D had significantly higher expression in T1DM rats compared to controls suggesting the potential involvement of NP pyroptosis in IVDD in T1DM rats [123]. Bmp7 was found to inhibit the activation of the Nlrp3 inflammasome and NP pyroptosis which mitigated IVDD in STZ induced T1DM rats [123]. Overall, results suggest Nlrp3 inflammasome activation and NP cell pyroptosis as likely contributors to IVDD and Bmp7 as a potential inhibitor of IVDD pathology in T1DM rats [123].

#### 4.3.3. Rabbits

New Zealand White Rabbits (*Oryctolagus cuniculus*) are commonly bred for research yet other breeds are also used. Rabbits are reported to reach skeletal maturity within 4–6 months of birth and most rabbits used for induced models of degeneration range from 1.5 to 9 months of age [154]. In these IVDD models, degeneration can typically be identified around four weeks post-surgical induction [154]. Because rabbits are commonly used species in induced IVDD models, quantitative analysis of degeneration is a useful tool in examination and interpretation of results. Sheldrick et al. found decay variance mapping of multi-echo transverse relaxation time (T2) MRI scanning on rabbits to be a beneficial quantitative analysis method with higher sensitivity and specificity than T2 relaxometry for disc puncture models [155].

Rabbits, similar to mice, retain NC-like cells at birth and for a large portion of their lives [156]. Kim et al. concluded that CC cells located in the central NP originate from the CEP [157]. Hematoxylin and eosin staining in combination with polarized light microscopy enabled visualization of recently formed fibrocartilage with collagen fibers showing double refraction and displayed consistent staining between the NP and CEP [157]. Rabbit annulus puncture models of degeneration revealed CEP remodeling with increased bone modeling and decreased small molecule transport, as concluded by lowered gadodiamide diffusion in the AF four weeks post puncture and twelve weeks in the NP [158].

### 4.4. Ungulates

#### 4.4.1. Cows

Holstein cattle (*Bos taurus*) is a popular breed in research. Developmentally, bovine IVDs, like other species, have a NC derived NP region, and a mesodermal sclerotome derived AF and CEP [9,14]. A NC sheath forms and thickens containing acid mucopolysaccharides which persists until around 45–50 days of gestation at which point the NC begins to break down [129]. NC cells within the NC sheath increase throughout embryonic development until the fetus reaches 30 mm in length at which cell numbers within the NC sheath decrease [129]. When the fetus reaches 55 mm resegmentation of the vertebrae and subsequent changes of the NC occur [129]. NC cells in the bovine IVD have a granular appearance due to enzymatic activity where secretory activity was recorded as early in development as 10 mm in length [129]. Considering developmental similarities, relative ease of obtaining tissue, and degenerative capacities, Holstein cattle continue to be utilized in the field.

Contributing to the growing understanding of IVD pathophysiology, experimental results from bovine RISH, fluorescent RISH and other transcriptional profiling have all further supported the IVD tissues as heterogenous cell populations and have enabled further tissue marker discovery including stemness markers supporting progenitor and stem-cell like qualities of some IVD cell groups (Table 1) [14,28,67,69,70]. This information supports the need for single cell level analysis of IVD cells. ScRNA seq of the bovine IVD has further provided evidence of stemness, suggestive of IVD progenitor cells [68]. Furthermore, analysis of a young, healthy and degenerating, aged bovine matrisome helped to clarify changes in protein expression over time [159]. This revealed increased protein expression of PRELP and FINC in degenerating, aging IVDs along with a high protein expression of Col12 and Col14 in healthy, young IVDs [159].

Beyond protein expression changes, bovine cell density diminishes significantly over time from conception onwards in both the AF and NP regions [160]. Between conception and four weeks of gestation, cell densities in the bovine AF and NP were recorded at 11,435 and 17,426 count/mm^2^ respectively [160]. This density drops dramatically, logging 1258 and 488 count/mm^2^ between birth and 1 week for AF and NP, and then down to 71 and 106 between count/mm^2^ five and ten years [160]. Figure 5 reflects a lower cell density in the bovine NP compared to AF after nuclear DAPI staining, showing cell arrangement in the IVD in pairs, strings, clusters, and singles similarly to the cellular arrangement reported in Bonnaire et al. [160]. Interestingly, the majority of clustered cells were in advanced degenerated tissue compared to moderate degenerated discs, trauma, acute nuclear prolapse, or chronic nuclear prolapse cells [160].

#### 4.4.2. Horses

Despite the historical importance of domesticated horses (*Equus callabus*) in agriculture and transportation and the financial interest in racing horses, characterization of equine IVDs has been a contradictory topic with great delays in the initial diagnosis of IVDD in horses compared to dogs or humans. Horses have seven cervical vertebrae, 18 thoracic, five to six lumbar, five sacral, and 15–21 coccygeal vertebrae [161] separated by IVDs. Warmblood horses have a documented CEP and fibrous AF, however literature describing both a presence and absence of NP and NC cells exists [162]. Histological studies in 2018 by Bergmann et al. confirmed equine NPs with high PG levels and a distinguishable TZ between NP and AF, while still noting the distinction between NP and AF in horses is not as clear as for humans or dogs [162]. Previous studies were unable to distinguish between the NP and AF and reported no NP cells in the equine thoracolumbar region [163] or the cervical region [164]. Because horses were reported lacking in NP cells, IVDD diagnosis was not prevalent until recent years; Kreuger et al. claimed the lack of clinical IVDD diagnosis in horses was due to failures differentiating between AF and NP tissues and subsequent challenges classifying a breakdown of NP degeneration and changes to cellular composition in combination with a thinner AF [165].

Even with the recently distinguished NP region, differences between equine and human IVDs continue to exist. Utilizing immunolabeling of KRT18, a NC marker, failed to identify any NC cells across the NP region in two fetal vertebral columns (youngest 45 days) and one adult vertebral column [11,161]. It is possible that NC cells are initially present and then disappear prior to E45 in development. A different histological study reports the formation of the neural tube and NC with subsequent segmentation of vertebrae by day 21 of gestation in horses, yet it does not specifically mention NC cells at any time point [166]. Further studies using a broader range of NC markers on horses between fertilization and E45 are recommended to determine the role of a NC in equine IVD development. ScRNA seq of equine IVDs to search for NC markers may offer additional information.

Limited cases of reported equine IVDD are typically a late-stage diagnosis with severe neurologic symptoms including pelvic limb ataxia as early stages may show only subtle signs of compression and pain [167]. In 2020, an eight-sample case study reported of cervical and cranial thoracic equine IVDD from 2008–2018 [168]. Clinical manifestation of IVDD presented as difficulty or inability to lower the neck to the ground for grazing, fixation or locking of the neck after grazing for five minutes, stiff neck posture, mild to severe ataxia, stumbling, and forelimb lameness [168]. Radiological markers of IVDD included reduction of IVD space, irregular margins of endplates, and irregular margins of vertebral bodies [168]. Bergmann’s 2022 study described 41 warmblood horses and applied a modified canine disc degeneration grading, adding vascularization of AF, for consistency in classification of equine IVDD [11]. Of the variables considered, positive correlation was found between degeneration and the total histological score, tearing of cervical AF and NP tearing [11]. This grading scale allows for consistent categorization of progression of IVDD in horses that will make future diagnosis easier. Histological scoring of disc degeneration considered: tears in AF, vascularization of NP, CC-like cell presence in the NP, NC cells present in NP and ECM staining of the NP by alcian blue and picrosirius red to view PGs and tears in the NP [11]. Table 3 outlines clinical presentations, radiographic findings, biochemical, and histological signs reported in equine IVDD. There is greater decrease in pentosidine levels in the caudal cervical region which could be explained by increased mechanical loading in this spinal region and therefore greater disc degeneration [11]. Due to the quadrupedal movement and longer neck compared to other species, loading and stresses on equine spine differ from other species.

While horses certainly are not immune to disc and spinal diseases, their diagnosis and subsequent treatment plans have been delayed due to challenges characterizing discs. Delays in diagnosis of equine spinal disorders are not unique to IVDD diagnosis. The first study reporting lumbosacral IVD protrusion in horses occurred in 2016, prior to that, disc protrusion was only reported in cervical and thoracic regions [165]. Ultrasound imaging enabled that report following identification of severe lumbar disc protrusion while histopathological analysis revealed inflammation of the dorsal root ganglia (DRG) spinal nerve root, and degeneration of the spinal cord [165]. Analysis of serum biomarkers in horses with chronic back pain found significant increases in serum levels of glial fibrillary acidic protein (GFAP), commonly used as an astrocytic glial cell marker, and phosphorylated neurofilament-H (pNF-H), a common damage marker for neurons and axons, compared to healthy controls, offering a potential biomarker for chronic back pain [169]. Further research utilizing larger equine study samples in combination with widespread use of diagnostic imaging to determine the prevalence of IVDD in symptomatic and asymptomatic horses would be beneficial.

**Table 3 vetsci-10-00540-t003:** Outlines the reported clinical manifestation, radiographic findings, notable biochemical changes, serum changes, and histological reports for equine IVDD based on published literature.

Common Region of Degeneration	Clinical Presentation	Radiographic Findings	Biochemical Markers	Serum Analysis	Histological
Cervical sections (161 cervical degenerated discs (DD): 56 thoracic DD: 48 lumbosacral DD) [162]	Spinal ataxia, more severe in pelvic limbs [170]	Collapse of disc space [171]	Increase in pentosidine in AF and NP [11]	Elevated creatine phosphokinase [171]	Fiber degeneration of white matter [171]
Caudo-cervical region showed significantly more severe degeneration compared to other regions. [162]	Limited range of motion in neck [11]	Endplate sclerosis [171]	Advanced glycation end product (AGE) crosslinking [11]	Serum glutamic-oxalocetictransaminase slightly elevated [171]	Poor myelin staining [171]
	Severe neck pain [11]	Disc protrusion [170,171]	Decrease in hydroxylysine moderate in AF, severe in NP [11]	Elevated blood urea nitrogen [171]	Necrosis of individual neurons in region of disc protrusion [171]
	Lameness [170]		Increased Collagen type I in NP [11]	Elevated venous pH values [171]	Scattering of microglial cells [171]
	Spasticity [11]		No change in glycosaminoglycans in NP [11]	CSF showed elevated protein content of xanthochromia [172]	Swollen axons [171]
	Dysmetria [11]			Increased GFAP serum levels [169]	Degeneration of spinal cord in area of disc protrusion [172]
	Normal cutaneous sensation and cranial nerve function [11]			Increased pNF-H serum levels [171]	
	Positive sway response [11]				
	Proprioceptive deficits [11]				

### 4.5. Cats

#### 4.5.1. Large Cats

A retrospective study of large cats (*Panthera*) analyzed 13 lions, 16 tigers, four leopards, one snow leopard and three jaguars kept in a zoo environment of which three lions, four tigers, and a leopard was diagnosed with IVDD [13]. In this group, clinical presentations included rear limb atrophy, ataxia, and paresis, as well as a reported decrease in activity and radiographic, histologic, and necroscopic analysis recorded decreased disc space, lesions, spondylosis, herniations and subsequent injury to spinal cords [13]. Generally, degenerative changes were more frequently discovered in lumbar regions compared to cervical or thoracic discs in these large cats [13]. Additionally, all the affected cats showed decreased appetite and weight loss leading up to their deaths which when reported in combination with ataxia, limb atrophy, paresis, or changes in activity should be considered as clinical indicators of potential degenerative spinal disease. Limited radiographic analysis in large felids leave the onset of degenerative changes unclear, however this type of analysis is recommended for diagnosis of IVDD [13].

#### 4.5.2. Small Cats

Case studies of two domesticated small cats (*Felis catus*) brought into veterinary clinics due to reduced appetite, changes in level of activity and changes to urination and constipation revealed the first reports of sacrococcygeal IVD protrusion [173]. Clinical findings included pain with lumbosacral palpation, effects on tail movement and extension of pelvic limbs, yet [173] both cases showed unremarkable serum biochemistry [173]. MRI analysis revealed protruded discs at the S3-Cd1 levels with both cases showing a hypointense NP, while non-affected discs had hyperintense NPs. Mild spondylolysis at the S3-Cd1 level was reported in one case, while both cases reported angulation of the spine and presumed spinal nerve root compression [173]. A 2001 study examining six cats between three and nine years of age with IVDD, identified through spontaneous disc extrusion, explored cerebrospinal fluid (CSF), serum biochemistry, radiographs, and histopathological analysis of material recovered during a hemilaminectomy ranging from T13-L6 across the six cats as a means to surgically decompress the region [174]. In this study clinical symptoms included progressive or acute paraparesis, paraplegia, absence of voluntary urination, changes to pelvic limb gait, back pain, and fecal incontinence. CSF analysis resulted in two cats with CSF blood contamination, and two with neutrophilic inflammation. Between one and three extradural compressive lesions were found in each cat. Mineralization was confirmed in two cats through histopathological analysis of the removed material while degeneration was confirmed in all six of the cats [174]. A 2020 case study examined L5-L6 intramedullary disc extrusion in a 10-year-old cat [175]. In this study clinical presentation included paraparesis affecting the right side more prominently, loss of bowel control and urination, pain with manipulation of the lumbar spine, and loss of tail movement. Computerized tomography (CT) imaging showed mineralization which was not confirmed in histologic analysis. CT analysis did show a decrease in IVD space while histologic analysis of surgically removed material confirmed NP degeneration. MRI analysis showed mild spinal cord swelling, reduction of NP volume, and intramedullary lesion [175]. Both Debreque et al. 2020 and Knipe et al. 2001 had cases with evidence of mineralization in imaging analysis that failed to be confirmed through histologic analysis, suggesting CT or radiographic imaging is not always reliable in diagnosing mineralization of discs.

### 4.6. Others

#### American Black Bear

In 2012, the first report of IVDD in an American black bear (*Ursus americanus*) was confirmed by MRI [12]. Clinical signs included rear limb paralysis, however due to winter torpor, it was not possible to determine for how long this was occurring. Following euthanasia due to poor prognosis, spondylosis and severe spinal cord compression were discovered [13].

## 5. Discussion

Understanding the developmental origins and cellular composition of the IVD is important to grasp its contributions to the structural support and functionality it provides to the spine. Recent developments in the omics fields now facilitate a better understanding of cellular heterogeneity and cell metabolic functions on a molecular level and can provide a basis for better diagnostic tools to identify early IVDD onset including timing, symptoms and biomarkers in animals, especially large breeds [27]. Importantly, scRNA seq has tremendously improved understanding of cell population dynamics between healthy and degenerated discs and the type of cells populating the region. Notably, scRNA seq further suggests progenitor cells and expression of stemness genes in cellular subpopulations. Further research into these cell population dynamics and signaling that contributes to a dwindling NC-like phenotype will likely improve our understanding of IVDD pathophysiology beyond human patients. This research has begun to identify an infiltration of immune cells participating in the inflammatory response in IVDD.

Broadly, the main tissue types and biochemical makeup of the IVD is consistent across the animal species discussed, however cellular differences exist in the presence and fate of NC cells throughout an animals’ lifespan. Notably, horses appear to lack NC cells already very early on in embryonic development, however other species dependent differences exist. While early understanding from morphological observations suggested the complete disappearance of NC cells quickly after the NC sheath broke down, scRNA seq identified subpopulations of cells expressing NC markers, suggesting that NC-like cells are present into adulthood in many species, including humans. This is important as NC cells support a different ECM makeup than other NP cells. The breakdown of the ECM, including loss of PG, changes in matrix synthesis, and disruption of a balance between anions and cations, contributes to degeneration [48]. Furthermore, these changes can lead to a decrease in disc pressure and disc height, contributing to increased stress on the AF and CEP [75]. Inflammation is not only a product of degeneration but also a catalyst for degeneration through the induction of inflammatory cascades, up regulation in pro-inflammatory cytokines, and increased cell death [85,87,93].

Recognizing clinical signs of IVDD on a molecular level is important for diagnosis across animals, especially for large animals were imaging based diagnostics is challenging. Clinical signs of IVDD in severe cases across species often reported limb ataxia, changes in appetite, limited range of motion or increased signs of pain with movement and proprioceptive changes [11,12,13,170]. The spinal region with the highest prevalence of IVDD and degenerative changes varies among species, likely due to changes in movement patterns, and it is important to consider how clinical manifestation may vary. Spinal degeneration is most likely to occur in the cervical spine of horses, the lumbar region for cats and humans, and the thoracolumbar region for dogs [66,71,143,144,162], which may influence certain clinical symptoms. More extensive and scoping case studies would be beneficial to identify broader trends in IVDD clinical manifestation across different animal species. Clinical symptoms such as loss of appetite, loss of bowel control, incontinence, decreased ambulation, and limb paralysis or paraparesis indicate a need for further testing to rule out or diagnose IVDD (Table 3). At this time, MRI imaging is the most useful diagnostic tool of IVDD, enabling visualization of changes to disc height, endplate irregularities, and disc protrusions, however it is not easily applicable to large animals [174]. In cases where MRI is not possible, testing for increased serum levels of IL6, ferritin, creatine phosphokinase, mCRP, GFAP, pNF-H, increased plasma levels of CCL5 and CXCL6, decreased plasma levels of GRB10, and elevated CSF levels of xanthochromia or clusterin may be used to support an IVDD diagnosis (Table 2) [107,114,115,118,124,169,171].

Mechanical stress and IVD acidification have been shown to decrease ECM protein synthesis, alter nutrient diffusion, and increase pro-inflammatory cytokines. Recording biochemical and physical changes to the IVD in degeneration is extremely useful in identifying potential therapeutic targets and biomarkers of both degeneration and tissue types. However, it may be the case that pH levels affect IVD cells to varying extents between species as described. Macrophages may offer therapeutic relief, as they are reversing the pro-inflammatory effects of TNFα or up regulate ECM proteins [100,101]. Interestingly IVDD, shares many degenerative and inflammatory aspects of disease with type 1 diabetes mellitus, disc herniation, and osteoarthritis including up regulation of CRP. Exploring IVDD through the lens of each disease, such as CRP levels or examination of heightened pyroptosis pathways, may help delineate degeneration mechanisms of IVDD.

## 6. Conclusions

Exposure to environmental, mechanical, and metabolic stress factors alongside genetic predisposition influences IVDD progression in many species. Technological advances in the imaging and “omics” fields can identify IVDD on a molecular and cellular level and enable novel diagnostic and therapeutic measures, especially for larger species.

## Figures and Tables

**Figure 1 vetsci-10-00540-f001:**
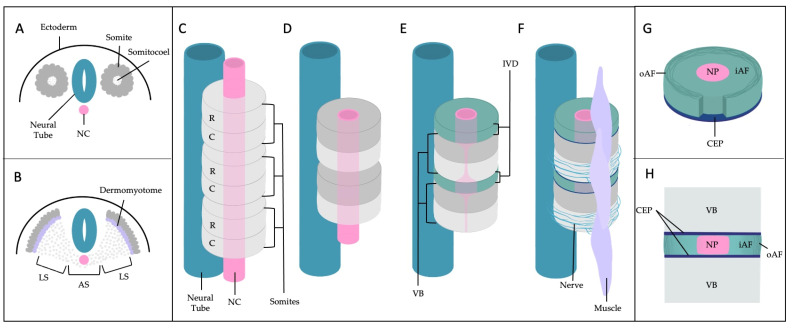
Simplified illustration of IVD formation adapted from [18,19,20,21,22,23,24]. (**A**) Specification of the embryonic axial skeleton, shortly after the formation of metameric somites is initiated by signals from the NC. (**B**) Ventromedial cells undergo an epithelial to mesenchymal transformation (EMT), taking on a mesenchymal sclerotome fate and migrate to surround the NC and neural tube. (**C**/**D**) Resegmentation of the axial sclerotome: The cell-dense caudal portion of one somite (C in (**C**)→ dark gray in (**D**)) will combine with the more loosely organized rostral portion of the adjacent somite (R in (**C**)→ light gray in (**D**)). (**E**) Following this resegmentation, the anterior layer of the cell dense section will give rise to the AF of the IVD (green), while the NP is derived from the NC (pink). Chondrogenesis enables the formation of the CEP and VB. (**F**) The reorganization and shift influenced by complex signaling events will enable innervation (light blue) of the myotome derived skeletal musculature (purple). (**G**) Transverse depiction of an isolated IVD illustrating only the caudal CEP for simplicity. (**H**) Depicts a sagittal section of the VB and IVD. The simplified illustration is not drawn to scale. AS: axial sclerotome; C: caudal; CEP: cartilage end plates; iAF: inner annulus fibrosus; IVD: intervertebral disc; LS: lateral sclerotome; NC: notochord; oAF: outer annulus fibrosus; R: rostral; VB: vertebral body.

**Figure 4 vetsci-10-00540-f004:**
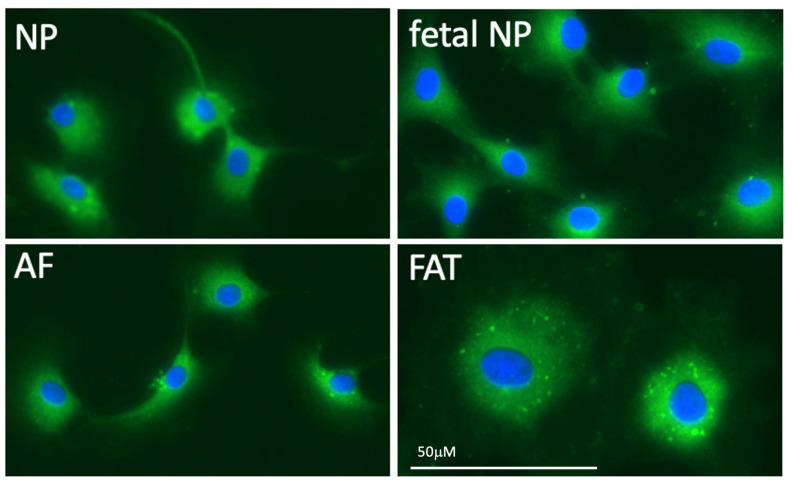
Fluorescent microscopy of primary adult coccygeal bovine IVD (NP and AF) and adipose (FAT) cells from the same animal and low passage fetal NP cells cultured in low glucose DMEM with 10% FBS. Cells were stained with MitoView^TM^ (green), which quickly accumulates in mitochondria. DAPI (blue) labels nuclei. All cells were stained and imaged the same day at the same magnification. Scale bar reflects 50 μm for all images. All tissue was collected as waste from local abattoirs.

**Figure 5 vetsci-10-00540-f005:**
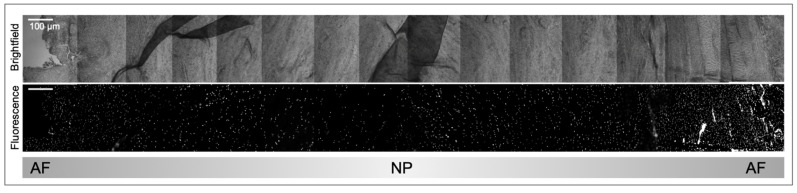
Top image shows a composite image created from overlayed sequential brightfield images of 7 μm paraffin sections of an adult bovine coccygeal IVD taken across the diameter of the disc from oAF to NP to oAF. The composite fluorescence image shows the DAPI labeled nuclei of each corresponding image. Scale bar represents 100 μm.

## Data Availability

Not applicable.

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
