# Peer review of "Development and Degeneration of the Intervertebral Disc—Insights from Across Species"

_vetsci, 2023, doi:10.3390/vetsci10090540_

Round 1

Reviewer 1 Report

-       The authors examine and present the mechanisms of intervertebral disk degeneration both within and between species. The in depth knowledge of this process is necessary for the prompt treatment and more over the prophylaxis of the disease.

-       The aims of their study are presented clearly and the areas of interest for further research are presented 

-       Although the search strategy of literature is not presented the structure of the manuscript is excellent helping us understand the search terms which are adequate for the paper

-       References that is essential for the arguments of the manuscript. The key statements are backed by references in all instances

-       Appropriate evidence is generally present – especially in horse and carnivore sections

-       Finally due to lack of side lines numbering a few comments are added as sticky notes.

Author Response

Thank you for reviewing our manuscript. Please find our responses in the uploaded PDF titled "Response cover letter 20230813".

Reviewer 2 Report

The article reads with great pleasure. The authors have described the   intervertebral disc (IVD) degeneration  in a unique way . They presented all possible aspects of IVD , describing it among other things from the point of view of structural and molecular changes. They presented research on IVD  in the following animal species: rats mouses, bovines, dogs, large and small cats, monkeys, rabbits, horses, and humans too. All chapters on the article are very interesting and properly presented. The 176 cited articles  indicate a very thorough analysis of the references. An article:"Development and Degeneration of the Intervertebral Disc – Insights from Across Species" is worth to be publish in Veterinary Sciences.

Author Response

(The authors gave the same response as above.)
